# FEATURE-GUIDED SCORE DIFFUSION FOR SAMPLING CONDITIONAL DENSITIES

## ABSTRACT

Score diffusion methods can learn probability densities from samples. The score of the noise-corrupted density is estimated using a deep neural network, which is then used to iteratively transport a Gaussian white noise density to a target density. Variants for conditional densities have been developed, but correct estimation of the corresponding scores is difficult. We avoid these difficulties by introducing an algorithm that guides the diffusion with a projected score. The projection pushes an image feature vector towards the corresponding centroid of the target class. The projected score and the feature vectors are represented and learned within the same network. Specifically, the image feature vector is defined as the spatial averages of the channels activations in select layers of the network. Optimizing the projected score for denoising loss encourages image feature vectors of each class to cluster around their centroids. It also leads to the separations of the centroids. We show that these centroids provide a low-dimensional Euclidean embedding of the class conditional densities. We demonstrate that the algorithm can generate high quality and diverse samples from the conditioning class. Conditional generation can be performed using feature vectors interpolated between those of the training set, demonstrating out-of-distribution generalization.

## 1 INTRODUCTION

Score diffusion is a powerful data generation methodology which operates by transporting white noise to a target distribution. When trained on samples drawn from different classes, it learns a mixture density over all the classes. In many applications, one wants to control the diffusion sampling process to obtain samples from the conditional distribution of a specified class. A brute force solution is to train a separate model on each class, learning each conditional density independently. This is computationally expensive: each model requires a large training set to avoid memorization (Somepalli et al., 2023; Carlini et al., 2023; Kadkhodaie et al., 2024). An alternative strategy is to train a single model on all classes, with a procedure to guide the transport toward the conditional density of individual classes. This approach can leverage the shared information between all classes, thus reducing the required training set size needed to learn the full set of conditional densities.

Learning conditional densities in a diffusion framework has been highly successful when the conditioning arises from a separately-trained text embedding system (e.g., Ramesh et al. (2021); Rombach et al. (2022); Saharia et al. (2022)) or image classifier network (Song et al., 2020; Dhariwal & Nichol, 2021), or by jointly learning a classifier and the score model Ho & Salimans (2022). Despite the high quality of generated images, several mathematical and numerical studies Chidambaram et al. (2024); Wu et al. (2024) show that these guidance algorithms do not sample from appropriate conditional distributions, even in the case of Gaussian mixtures. This is due to their reliance on estimating the exact likelihood to obtain the score of the conditional distributions, which is difficult.

In this work, we introduce a modified score diffusion, which does not rely on direct estimation of the score of conditional densities. Instead, at each step of the trajectory, it modifies the score according to the distance between the sample and the target conditional distribution in a feature space. Importantly, the score and the feature vector are represented by the same neural network learned by minimizing a single denoising loss. The feature vector is defined as spatial averages of selected layers of the score network. This shared representation provides a Euclidean embedding of all class

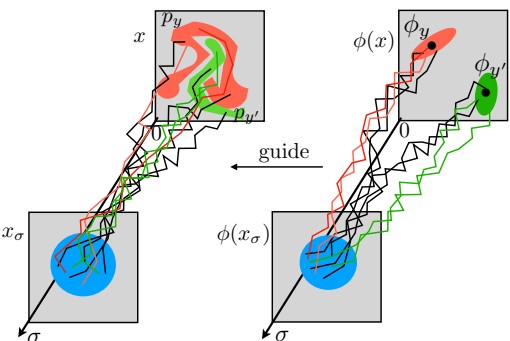

Figure 1: Illustration of feature-guided score diffusion. Left: Score diffusion of a mixture of densities computes trajectories (black) that map samples of a Gaussian white noise (blue disk) to samples of two complex conditional densities (orange or green). Right: The feature space $\phi(x)$ defines a Euclidean embedding in which each mixture component is well separated (orange/green ellipses). In the embedding space, mixture trajectories (black) are similar at high noise variance $\sigma^2$, and bifurcate, moving toward different components at lower noise levels (Biroli et al., 2024). In our method, feature trajectories (orange/green) are forced toward the feature centroids ($\phi_y$ or $\phi_{y'}$, on right) of the corresponding conditional density ($p_y$ or $p_{y'}$, on left). These feature trajectories are used to guide the trajectories of $x_\sigma$ in the signal space (orange/green, left) toward the corresponding conditionals.

conditional probabilities. The sampling algorithm relies on this Euclidean embedding to sample from the conditional density.

Several methods have have been developed to learn representations in conjunction with diffusion models (Preechakul et al., 2022; Mittal et al., 2023; Wang et al., 2023; Hudson et al., 2024). In general, these models use a separate network to map images into a form that can be used to control a diffusion network. Training these models can be difficult, due to mixed-network architectures, and use of objective functions with combined denoising and other losses. Although they have proven successful, in terms of image generation quality, or to transfer of the learned representation to other tasks, the Euclidean metric of the embedded space has not been related to properties of underlying probability distributions.

In this work we propose an algorithm that samples a class conditional density by guiding the score diffusion with a feature vector, which is driven towards the class centroid in the feature space. This is illustrated in Figure 1. It computes a projected score at each step such that the trajectory samples from the conditional density. We show numerically that the learned features concentrate in the neighborhood of their centroid within each class. We also verify that the centroids of feature vectors define a Euclidean embedding of the associated conditional probability, and are thus separated according to the distance between these conditional distributions. As a result, we find that this feature-guided sampling procedure can accurately sample from the target conditional probability density, without degradation of quality or diversity. Both training and synthesis are stable. We show that for Gaussian mixtures, the method recovers distributions which closely match each Gaussian component. Finally, we demonstrate that the Euclidean embedding allows sampling of conditional probabilities over new classes obtained by a linear combination of the feature vectors of two classes.

## 2 BACKGROUND

**Sampling by score diffusion.** Sampling using score diffusion (more precisely, *reverse diffusion*) is computed by reversing time in an Ornstein-Uhlenbeck equation, initialised with a sample $x$ drawn from probability density $p(x)$. At each time $t$ the diffusion process computes a noisy $x_t$ with a Gaussian probability density $p_t = \mathcal{N}(e^{-t}x, \sigma_t^2 I)$. At large time $T$, $x_T$ is nearly a Gaussian white noise. One can recover $x$ from $x_T$ by reversing time $T$ to 0 using a damped Langevin equation:

$$-dx_t = \big(x_t + 2s(x_t)\big)dt + \sqrt{2}dw_t \qquad (1)$$

where $s$ is a drift term and $w_t$ is a Brownian noise. If $s(x_t) = \nabla_{x_t} \log p_t(x_t)$ is the score of $p_t$ then this score diffusion equation transports Gaussian white noise samples into samples of $p$. To

implement a score diffusion, the main difficulty is to estimate the score $\nabla_{x_t} \log p_t$. However, there is a considerable freedom to choose the drift term $s(x_t)$ (Albergo M., 2023). We will later leverage this degree of freedom.

The score is typically estimated by minimizing a mean squared error denoising loss. To specify the denoising problem, we renormalise $x_t$ and define $x_\sigma = e^t x_t$, whose probabilty density $p_\sigma$ is parametrised by $\sigma = e^{2t} - 1$. The denoising solution provides a direct constraint on the score, $\nabla_{x_\sigma} \log p_\sigma(x_\sigma)$, thanks to a remarkable formula derived by Tweedie (as reported in Robbins (1956)) and Miyasawa (1961):

$$\hat{x}(x_\sigma) = \mathbb{E}[x|x_\sigma] = x_\sigma + \sigma^2 \nabla_{x_\sigma} \log p_\sigma(x_\sigma) \tag{2}$$

The score can be estimated with a neural network that computes a function $s_\theta(x_\sigma)$ whose parameters are chosen to minimize a denoising loss (Song & Ermon, 2019; Ho et al., 2020):

$$\ell(\theta) = \mathbb{E}\|s_\theta(x_\sigma) - \sigma z\|^2 = \mathbb{E}\|x - \hat{x}(x_\sigma)\|^2. \tag{3}$$

**Conditional sampling.** Suppose that we have a dataset of independent samples $\{x_i, y_i\}_{i \le n}$ where $x_i$ is an image and $y_i$ is a label which may correspond to a discrete class or a continous attribute. These are samples of a probability density that is a mixture of conditional densities: $p(x) = \int p_y(x) \, p(y) dy$, where $p_y(x) = p(x|y)$ is the conditional probability of $x$ given $y$, and hence of the samples of class $y$. Let $p_{y,\sigma}(x_\sigma) = p_\sigma(x_\sigma|y)$ be the probability density of noisy samples at a certain noise level $x_\sigma = x + \sigma z$ over all $x$ in class $y$ and $z \sim \mathcal{N}(0, Id)$. Samples of $p_y$ can be generated using a score diffusion algorithm if one has estimates of the scores $\nabla_{x_\sigma} \log p_{y,\sigma}(x_\sigma)$ *for all* $\sigma$. Bayes' rule gives

$$\nabla_{x_\sigma} \log p_\sigma(x_\sigma|y) = \nabla_{x_\sigma} \log p_\sigma(y|x_\sigma) + \nabla_{x_\sigma} \log p_\sigma(x_\sigma). \tag{4}$$

It is thus tempting to use this equation to compute the conditional score by augmenting the unconditioned score (second term on right) with an estimate of the gradient of the log-likelihood (first term). This approach relies on estimating the likelihood from data *at all noise levels*. In practice, one might employ neural network classifiers trained on clean data to estimate the likelihood. This however introduces an error because the likelihood function depends on noise level, and thus $p_\sigma(y|x_\sigma) \ne p(y|x_\sigma)$. As a result, the correct likelihood of $y$ for noisy data cannot be computed by evaluating the likelihood function $p(y|.)$ on noisy data. This problem has been addressed by training classifiers on noisy data. However, in practice, obtaining a good estimation of the likelihood at all noise levels has been challenging. In particular, samples drawn using this algorithm are low in quality and often do not belong to the correct class Dhariwal & Nichol (2021). This problem persists with classifier-free guidance Ho & Salimans (2022), where the likelihood gradient is computed with the score network, and a weight $\omega > 0$ is chosen to emphasize the log-likelihood term $(1 + \omega)\nabla_{x_\sigma} \log p(y|x_\sigma) + \nabla_{x_\sigma} \log p_\sigma(x_\sigma)$. Such algorithms generate high quality images Dhariwal & Nichol (2021); Ho & Salimans (2022) but they do not correctly sample the conditional distribution, drawing instead from a desnity of reduced diversity. Even in the simplified case of Gaussian mixtures, the conditional density errors are significant, as proven and demonstrated numerically in Chidambaram et al. (2024); Bradley & Nakkiran (2024).

## 3 FEATURE-GUIDED SCORE DIFFUSION

We present a method for learning and sampling from conditional distributions without direct likelihood estimation. Instead, we augment the score of the mixture distribution with a projection term that operates over learned feature vectors, that serves to push diffusion trajectories toward the density of the desired conditional distribution.

**Trajectory dynamics for a Gaussian mixture.** The dynamics of score diffusion for mixture of densities has been studied in Biroli et al. (2024). When the underlying $p(x)$ is simply a mixture of Gaussians with low rank covariance, the score diffusion of this mixture can be described in roughly three phases as illustrated in Figure 1. Initially, $\sigma$ is large and $x_\sigma$ is dominated by the Gaussian white noise, so its probability distribution is nearly Gaussian and trajectories are nearly identical for all classes $y$. At some noise variance, which is dependent on the distance between the means of the mixture components, the density becomes multi-modal and the trajectories separate. Once trajectories are separated, they fall into the basin of attraction of a single component density $p_y$ and converge to

samples of $p_y$. In the third stage, when the noise is sufficiently small, $\nabla_{x_\sigma} \log p_\sigma \approx \nabla_{x_\sigma} \log p_{\sigma,y}$, because the other components have a negligible effect on $\nabla_{x_\sigma} \log p_\sigma$.

To sample conditional densities, we must control the trajectory so that it is pushed toward the basin of attraction of $p_y$ at all noise levels. This can be done by adding a forcing term to the mixture score $\nabla_{x_\sigma} \log p_\sigma$. Consider the simple case of a Gaussian mixture $p(x) \propto (e^{-(x-m_1)^2/(2\lambda^2)} + e^{-(x-m_2)^2/(2\lambda^2)})$ with means $m_2 = -m_1$, with $\lambda^2 \ll |m_1|^2$. To approximately sample $p_y$, an adjusted score may be defined with a forcing term proportional to $m_y - x_\sigma$:

$$s(x_\sigma, m_y - x_\sigma) = \nabla_{x_\sigma} \log p_\sigma(x_\sigma) + K_\sigma (m_y - x_\sigma).$$

This is the gradient of the log of $p_\sigma(x_\sigma) e^{-K_\sigma(x_\sigma-m_y)^2/2}$, which drives the transport toward the mean $m_y$. To sample from the component with mean $m_y$, $K_\sigma$ must be sufficiently large at high noise variance $\sigma^2$ to drive the dynamics to $m_y$. It must then converge to zero for small $\sigma^2$, so that the modified score diffusion samples a distribution which is nearly a Gaussian with variance $\lambda^2$.

**Feature concentration and separation.** The Gaussian mixture example provides inspiration for sampling from mixtures of complex distributions $p_y$. In the Gaussian case, the linear forcing term can be defined in terms of the class means $m_y = \mathbb{E}_{p_y}[x]$, because the component distributions for each $y$ are sufficiently concentrated around their means to be well-separated. For mixtures of complex distributions, to apply a similar strategy we must find a feature map $\phi(x)$ such that the mapped conditional distributions for each $y$ concentrate around their corresponding means $\phi_y = \mathbb{E}_{p_y}[\phi(x)]$. Moreover, all $\phi_y$ must be sufficiently *separated*. This is obtained by insuring that $\phi_y - \phi(x)$ for $x$ in class $y$ has relatively small projections in the directions of all $\phi_y - \phi_{y'}$:

$$\forall y, y' \;\;,\;\; \mathbb{E}_{p_y}[\langle \phi_y - \phi_{y'}, \phi_y - \phi(x) \rangle^2] \ll \|\phi_y - \phi_{y'}\|^2. \tag{5}$$

The separation of $\phi_y$ in the embedding space should be governed by the separation of the probability distributions $p_y$ in the pixel space. This is captured by a Euclidean embedding property, which ensures that the separation of $\phi_y$ is related to a distance between the probability distributions $p_y$, and hence that there exists $0 < A \leq B$ with $B/A$ not too large, such that

$$\forall y, y' \;\;,\;\; A\|\phi_y - \phi_{y'}\|^2 \leq d^2(p_y, p_{y'}) \leq B\|\phi_y - \phi_{y'}\|^2. \tag{6}$$

Since $\phi_y - \phi(x_\sigma)$ must control $\nabla_{x_\sigma} \log p_{y,\sigma}$ at all noise levels, we establish a distance between two conditional densities as

$$d^2(p_y, p_{y'}) = \int_0^\infty \Big( \mathbb{E}_{p_{\sigma,y}}[\|\nabla_{x_\sigma} \log p_{\sigma,y}(x_\sigma) - \nabla_{x_\sigma} \log p_{\sigma,y'}(x_\sigma)\|^2]$$
$$+ \mathbb{E}_{p_{\sigma,y'}}[\|\nabla_{x_\sigma} \log p_{\sigma,y}(x_\sigma) - \nabla_{x_\sigma} \log p_{\sigma,y'}(x_\sigma)\|^2] \Big) \sigma \, d\sigma. \tag{7}$$

This distance is based on the difference in the expected score assigned to $x_\sigma$ by $p_y$ vs. $p_{y'}$, integrated across all noise levels. It provide a distance by symmetrizing the Kullback-Leibler divergence $KL(p_y \| p_{y'})$ between two distributions $p_y, p_{y'}$ proved in (Song et al., 2020):

$$KL(p \| p') = \int_0^\infty \mathbb{E}_{p_\sigma}[\|\nabla_{x_\sigma} \log p_\sigma(x_\sigma) - \nabla_{x_\sigma} \log p'_\sigma(x_\sigma)\|^2] \sigma d\sigma.$$

The feature concentration and separation properties can also be reinterpreted as an optimization of a nearest mean classifier

$$\hat{y}(x) = \arg\min_y \|\phi(x) - \phi_y\|^2.$$

In that sense, the control of the score by $\phi_y - \phi(x_\sigma)$ is related to classifier-guided score diffusion (Dhariwal & Nichol, 2021).

**Projected score.** In the Gaussian mixture case, we described an augmentation of the score with a forcing term that is linear in the deviation $m_y - x_\sigma$. For mixtures of complex probability distributions, we choose to adjust the score with an analogous forcing term that operates in the embedding space: $e = \phi_y - \phi(x_\sigma)$. We define $\phi$ using the activations within the same neural network that computes the score, which allows us to make use of nonlinear representational properties of the score network (Xiang et al., 2023), and to jointly optimise $s$ and $\phi$. This shared parameterization is crucial to

ensure that the embedding arises from the same features that represent the score of the conditional distribution, which in turn renders the embedding space Euclidean in relation to the probability space. Specifically, for images, the components of $\phi(x_\sigma)$ are defined as spatial averages of activations of a selected subset of layers of the deep neural network that computes $s(x_\sigma)$ (see Appendix A for architecture diagram). Activation layer averages are close to the first principal component of the network channels whose values are all positive, thus capturing a significant fraction of their variance. This feature vector is translation-invariant (apart from boundary handling), and has far fewer dimensions than the image. It can thus be considered a bottleneck. The fact that the same network is used to compute $s$ and $\phi$ is a critical aspect of our algorithm which sets it apart from previous approaches for score-based representation learning.

We define $s(x_\sigma, \phi_y - \phi(x_\sigma))$ by multiplying each component of $\phi_y - \phi(x_\sigma)$ with a learned factor and adding it to the corresponding activation layers of $s(x_\sigma, 0)$. For multiplicative factors smaller than 2, the selected activation layers of $s(x_\sigma, \phi_y - \phi(x_\sigma))$ have an average closer to $\phi_y$. Learned factors are often close to 1, which sets averages to $\phi_y$. In this case, the operation can be interpreted as a projection in the embedding space, and thus we refer to $s(x_\sigma, \phi_y - \phi(x_\sigma))$ as the *projected score*. This procedure, of matching the means of channels to those associated with a target density, is inspired by methods used for texture modeling and synthesis (Portilla & Simoncelli, 2000).

At high noise levels this projection or contraction drives the dynamics toward the class $y$, as shown in Figure 1. In this regime, $\phi(x_\sigma)$ has significant fluctuations which carry little information about $x$. Projecting it to $\phi_y$ reduces these fluctuation and uses the conditioning information to push the transport toward $p_y$. At the final steps of the dynamics, when the noise level is small, we have $x_\sigma \approx x$. The concentration property of $\phi(x)$ implies that deviation $e = \phi_y - \phi(x_\sigma)$ is small. At small noise levels, the dynamics conditioned by $y$ should follow nearly the same dynamics as the mixture, and thus $s(x_\sigma, 0) \approx \nabla_{x_\sigma} \log p_\sigma(x_\sigma)$. A first order approximation of $s(x_\sigma, e)$ relative to $e$ gives

$$s\big(x_\sigma, \phi_y - \phi(x_\sigma)\big) \approx \nabla_{x_\sigma} \log p_\sigma(x_\sigma) + \big(\phi_y - \phi(x_\sigma)\big)^T \nabla_e s(x_\sigma, e)|_{e=0}. \tag{8}$$

At small noise levels, the projected score is thus approximated by the unconditioned mixture score with a forcing term that is linear in the feature deviation $\phi_y - \phi(x_\sigma)$. This projected score is the basis for our feature-guided score diffusion algorithm, which is implemented using Stochastic Iterative Score Ascent (SISA) (Kadkhodaie & Simoncelli, 2021) (see Appendix C and Algorithm 1).

## 4 JOINT LEARNING OF FEATURES AND PROJECTED SCORE

Learning $s(x_\sigma, \phi_y - \phi(x_\sigma))$ by minimizing a denoising loss over all $y$ and $x$ does not ensure that $\phi(x)$ concentrates within class $y$, because this property is not explicitly imposed. It can however be encouraged by replacing $\phi_y$ with $\phi(x')$ in the learning phase, where $x'$ is a randomly chosen sample from the same class $y$ as $x$. The learning algorithm thus optimizes the parameter $\theta$ of a single network $s_\theta(x_\sigma, \phi_\theta(x') - \phi_\theta(x_\sigma))$ for randomly chosen $x'$, by minimizing the denoising loss

$$\ell(\theta) = \mathbb{E}_{x, x', \sigma} \| z - s_\theta\big(x_\sigma, \phi_\theta(x') - \phi_\theta(x_\sigma)\big) \|^2,$$

where the expected value is taken over the distribution of all $x$ in the mixture, over all $x'$ in the same class as $x$, and over all noise variances $\sigma^2$. Note that both the projected score, $s$, and the feature vector, $\phi$, are dependent on the network parameters $\theta$, and are thus simultaneously optimized. See Algorithm 2 of Appendix C for more details.

**Qualitative analysis of denoising optimization.** We provide an intuition for how minimizing denoising loss interacts with $\phi_\theta(x') - \phi_\theta(x_\sigma)$ inside $s_\theta$ to learn the desired projected score. Specifically, we give a qualitative explanation for why minimization of the denoising loss encourages a feature vector $\phi$ that concentrates in each class and has separated class means $\phi_y$. Concentration is a consequence of optimization at small noise and separation is due to optimization at high noise levels.

At sufficiently small noise, when $x_\sigma \approx x$, and $x_\sigma$ is in the basin of attraction of $p_y$, the projected score should converge to the score of the mixture model

$$s_\theta(x_\sigma, \phi_\theta(x') - \phi_\theta(x_\sigma)) \approx \nabla_{x_\sigma} \log p_\sigma(x_\sigma).$$

So deviation from the score of the mixture model is tantamount to an increase in loss. Thus, to minimize the loss, the parameters of the network are learned such that at small noise levels $\phi_\theta(x') - \phi_\theta(x_\sigma)$ becomes very small for all pairs in the class, hence convergence of $\phi(x)$.

The convergence of feature vectors within classes does not guarantee separations of their centroids. This is a major challenge in representation learning known as "collapse". This pathological case is avoided thanks to loss minimization at high noise levels. This is a regime where conditioning can reduce the loss below the mixture model loss. The high level of noise obfuscates image features such that $x_\sigma$ becomes high probability under classes other than $y$. So, if loss minimization results in $\theta$ such that $\phi(x')$ approximates $\phi_y$, projected score leads to a better estimate of $x$, hence a lower denoising loss. Therefore, at high noise we expect improved performance relative to the mixture distribution, whereas at small noise we expect to achieve similar performance. This requires that the $\phi_y$ of different classes have a separation of the order of the separation of conditional densities. The separation of the $\phi_y$ thus depends on the separations of the $p_y$. It leads the optimization to define $\phi_y$ providing a Euclidean embedding of the $p_y$.

## 5 EXPERIMENTAL RESULTS

We trained a UNet on cropped $80 \times 80$ patches from a dataset of 1700 texture images following Algorithm 2 (see Appendix A for details of architecture and dataset). The feature vector consists of spatially averaged responses of layers at the end of each block, at all levels of the U-Net, which correspond to different scales. The full feature vector has 1344 components. Patches from each image are assumed to represent samples of the same class. Each training example consists of one noise-corrupted patch, $x_\sigma$, and another patch that is used to compute an embedding vector for conditioning, $\phi(x')$. The UNet implementation has receptive field (RF) of size $84 \times 84$ at the last layer of the middle block, ensuring that $\phi$ can represent global features of the patches. We also used Algorithm 4 to train a UNet of identical architecture to denoise patches, representing the full mixture density without conditioning. We refer to this as the "mixture denoiser".

### 5.1 PROJECTED SCORE IMPROVES DENOISING

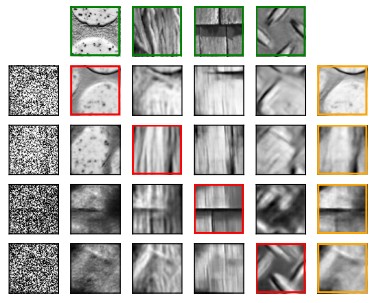 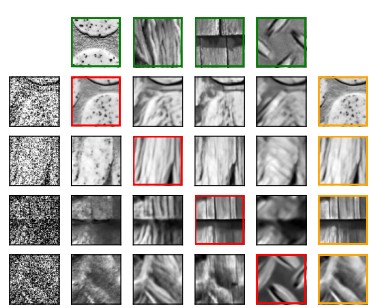

Figure 2: Feature guided denoising results at two noise levels (left: $\sigma = 1$, right: $\sigma = 0.5$). Leftmost column of each panel shows noisy images, drawn from 4 classes. Top row (green boxes) shows example conditioning images, from the same 4 classes. Columns under each show corresponding denoising results. Diagonal entries (red boxes) indicate images denoised with correct conditioning (conditioning image from same class as noisy image), whereas off-diagonal entries are incorrectly conditioned. Rightmost column of each panel shows denoising results using the (unconditioned) mixture denoiser (orange boxes). At high noise levels, conditioning on the correct class improves results significantly compared to the mixture model. Conditioning on the wrong class degrades performance, introducing features from the conditioning class. At smaller noise levels, feature guided and mixture denoisers produce similar outputs, but the effect of incorrect conditioning is still visible.

We first evaluated denoising performance of the feature guided denoiser, to verify the analyses and predictions of Section 3. Example denoising results for four different image classes, and two different noise levels are shown in Figure 2. In all cases, feature guidance has a visually striking effect, pushing the denoised images toward the conditioning class. These effects are more substantial at the higher noise level, as predicted from the analyses of Section 3. Moreover, performance is substantially worsened by incorrect conditioning (i.e., denoising an image drawn from $p(x|y_i)$, while conditioning on feature vector $\phi_{y_j}$, with $i \neq j$). In these cases, deformations and artifacts in the denoised images resemble prominent features of the (incorrect) conditioning class. A quantitative comparison of

denoising performance is shown in Figure 3(left), and further supports the predictions of Section 3. At all noise levels, conditioning improves performance. However, as predicted by Equation (8), this improvement decreases monotonically with noise level, because the projected score converges to the original mixture score. At the smallest noise level, the two models have nearly identical performance.

In Figure 3(right) we compare performance to a denoiser optimized for a single class. This model uses a UNet with identical architecture, and is trained on 125000 crops from one texture class $y_0$ using Algorithm 2 (see Appendix A for details of dataset). This model provides an empirical upper bound on the denoising performance, and hence the conditional score, $p(x|y_0)$, for class $y_0$. The results indicate that the feature guided denoiser gets close to but falls short of exactly achieving the best empirically possible conditional score for this architecture, as anticipated in Section 4. On the other hand, the feature guided model is better than the single-class denoiser when conditioned on the wrong class. Despite this suboptimality in approximating the true conditional score, we show in the Section 5.3 that the feature guided denoiser can nevertheless be used to draw diverse high-quality samples from class-conditioned densities.

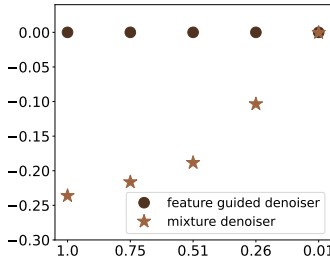 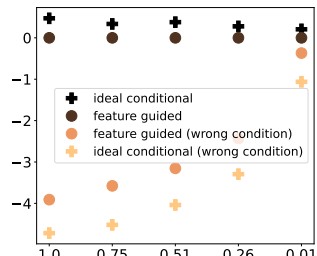

Figure 3: **Left:** Improvement in peak signal to noise ratio (PSNR) at different noise levels, of the conditional model (discs) relative to the unconditioned mixture model (stars), averaged over samples from all classes. **Right:** Comparison of conditional model (discs) with a denoiser optimized for a single class $y_0$ (stars). Upper points correspond to denoising of images from class $y_0$, with correct conditioning. Lower points correspond to denoising of images from other classes, $y \neq y_0$, with incorrect conditioning.

## 5.2 PROPERTIES OF LEARNED EMBEDDING

We verify the concentration, separation and Euclidean embedding properties of feature vectors which are needed to guide the score diffusion. Figure 4 shows the squared Euclidean distance between feature vectors of images drawn from the same class, and for the mean feature vectors from different classes. The top row is computed for the (unconditioned) mixture network. Note that the feature vectors are highly concentrated, and there is some moderate separation between classes, consistent with Xiang et al. (2023). The bottom row shows the same results for feature guided model. The histogram of variances of feature vectors within classes is more concentrated, and overlaps less with the Euclidean distances between class feature vectors, in comparison with the mixture model. Thus, the feature guided model exhibits stronger concentration and separation in the embedding space. We also examined these properties over different stages of the UNet. The middle column of Figure 4 shows that the separation between the class centroids is most significant in the middle layer of the network. In this block, the network receptive field size is as large as the input image, enabling it to capture global features that are most useful for separating classes. This effect is shown for one pair of classes in the right column.

We can also verify that the learned class feature vectors provide a Euclidean embedding of the conditional probabilities. Figure 5 shows a scatterplot of the density distance $d^2(p_y, p_{y'})$ (Equation (7)) as a function of the Euclidean distance, $\|\phi_y - \phi_{y'}\|^2$, over pairs of different classes $\{y, y'\}$ in the embedding space. The data are well-approximated by a line, satisfying the conditions of Equation (6) for reasonable $A, B$.

## 5.3 CONDITIONAL GENERATION

Finally, we evaluate the numerical performance of feature guided score diffusion to sample conditional probabilities of Gaussian mixtures and mixtures of image classes.

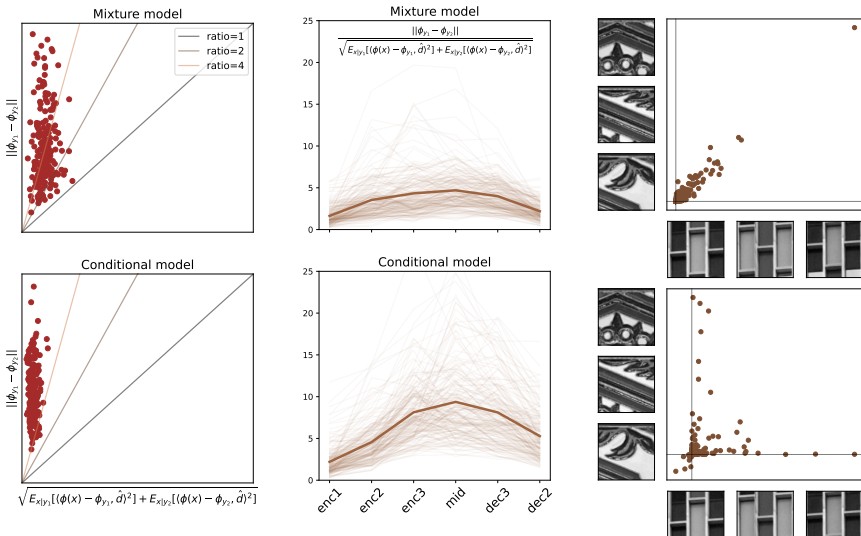

Figure 4: Concentration of image feature vectors $\phi(x)$ within class, and separation of centroids $\phi_y$ between classes. Top row shows results for the unconditioned mixture model and the bottom row shows results for the conditional model. **Left column:** scatter plot of Euclidean distances between pairs of class feature vectors $\phi_{y_1}$ and $\phi_{y_2}$ versus the average variability of feature vectors within classes. $\hat{d}$ is the unit vector in the direction of $\phi_{y_1} - \phi_{y_2}$ (i.e., the *prototype* classifier). Separability of classes corresponds to the ratio of the two coordinates (lines provide reference for three specific examples). **Middle column:** Separability of class feature vectors (ratio of distances between class feature vectors to average variability within the classes) computed for portions of the feature vectors corresponding to different layers of the UNet architecture. The conditional model separates classes significantly, especially in the middle layer. **Right column:** scatter plot of components of $\phi$ for two different classes, $y_1$ and $y_2$, in the middle layer. Example images from $y_1$ and $y_2$ are shown along the axes. The image embeddings in the conditional model are separated, while there is very little separation in the mixture model.

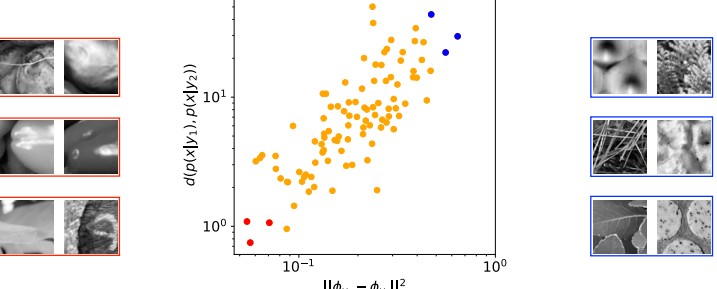

Figure 5: Verification of Euclidean embedding (Equation (6)). Density distance (Equation (7)), which bounds the symmetrized KL divergence between the two conditional densities, is well-correlated with the squared Euclidean distance between the corresponding mean feature vectors in the embedding space. Image pairs on the left are drawn from the closest three class pairs (red points), and those on the right are drawn from the most distant (blue points).

**Gaussian mixtures.** Guided diffusion models (Ho & Salimans, 2022) have been highly successful in generating text-conditioned images, but recent results demonstrate that they do not sample from the conditional density on which they are trained. This is proved for mixtures of two Gaussians (Chidambaram et al., 2024), which captures important properties of the problem. We trained our model on samples from such a two-Gaussian mixture, having different means $m_1$ and $m_2$ and a rank 1 covariance whose principal component is $(m_2 - m_1)/\|m_2 - m_1\|$. Figure 6 shows the distribution of

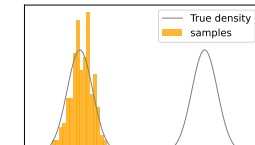 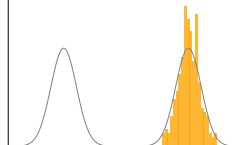

Figure 6: Conditional sampling for a mixture of two Gaussians. Network is trained on samples from the mixture, and the two panels show histograms (yellow) of samples drawn conditioned on each of the classes.

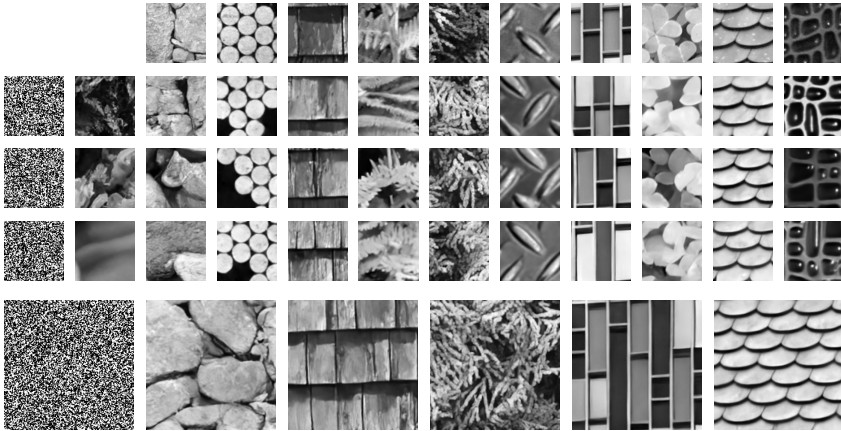

Figure 7: Conditional sampling. Top row shows example images from different conditioning classes $y$. Leftmost column shows initial (seed) noise images. Second column (3 small images only) shows samples from the (unconditioned) mixture denoiser, trained on all classes. Remaining columns show 3 images sampled using the conditional model, conditioned on the feature vector $\phi_y$ for the corresponding class. Here the class feature vector is obtained from a single image from a class (i.e., setting $n_y = 1$ in Algorithm 1). Bottom row shows larger synthesized images, each sampled conditionally from the class corresponding to the leftmost of the two columns above it, and initialized by the noise image on the left.

conditional samples generated by Algorithm 1. Our feature guided score diffusion generates typical samples from each Gaussian conditional density.

**Natural images.** We trained a network on pairs of $80 \times 80$ patches selected randomly from a dataset of 1700 grayscale texture images (i.e. 1700 classes). We generated samples by using the trained model in Algorithm 1. Figure 7 shows three samples generated for each of 10 different classes, as specified by their corresponding feature vectors $\phi_y$. Samples are visually diverse, of high quality, and appropriate for the corresponding conditioning class. The bottom row shows samples drawn at twice the resolution, using 5 of the same conditioning classes. Figure 10 and Figure 11 in Appendix D show more examples of conditional sampling. Additionally, Figure 12 and Figure 13 show the effect on conditioning at different noise level on sampling.

Figure 8 demonstrates that interpolation within the embedding space is well-behaved. Each row shows samples using a conditioning vector in the embedding space that is interpolated between those of two classes, $\{y_1, y_2\}$. The rows are ordered by the Euclidean distance between the class feature vectors, $\|\phi_{y_1} - \phi_{y_2}\|$. In all cases, the generated samples are generally of high visual quality, and represent a qualitatively sensible progression.

## 6 DISCUSSION

We presented a feature guided score diffusion method for learning a family of conditional densities from samples. A projected score guides the diffusion in a feature space where the conditional densities are concentrated and separated. Both the projected score and the feature vectors are computed on internal responses of a deep neural network that is trained to minimize a single denoising loss. When conditioned on the feature vector associated with a target class, a reverse diffusion sampling

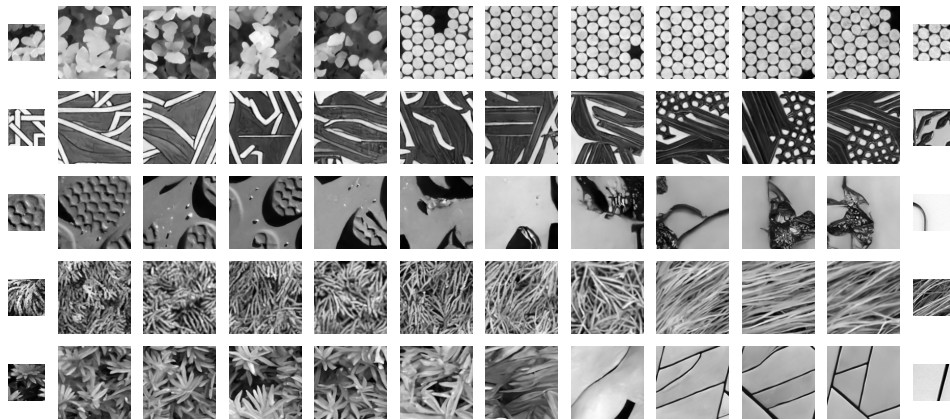

Figure 8: Interpolation in embedding space. Each row shows high-resolution samples drawn from $p(x|\alpha\phi_{y_1} + (1 - \alpha)\phi_{y_2})$ for different class pairs $\{y_1, y_2\}$, with representative samples from the training set shown on left and right sides. Rows from top to bottom correspond to pairs of classes with increasing Euclidean embedding distance.

algorithm based on the projected score transports a Gaussian white noise density to the target conditional probability following a trajectory that differs from that of the true conditional score. We demonstrate this numerically by showing that denoising performance remains below that of the optimal conditional denoiser. Nevertheless, a diffusion algorithm based on the projected score provides an accurate sampling of conditional probabilities, which is demonstrated for Gaussian mixtures and by testing the quality and diversity of synthesized images. We also verify that the feature map provides a Euclidean embedding of corresponding conditional probabilities, which allows us to interpolate linearly between classes in the feature space.

Our method is novel, but bears some similarity to several others in recent literature, each of which aim to learn a density (or at least, a diffusion sampler) conditioned on an exemplar from a class. Most of these are significantly more complex to train than our network, relying on multiple interacting networks, often with multiple-term objectives. Ho & Salimans (2022) introduced classifier-free guidance, in which the score estimates of a conditional diffusion model are mixed with those of an unconditioned diffusion model. They were able to obtain high-quality samples, but the likelihood term in the conditional model over-biased the conditional sampling, resulting in a mismatch to the conditional density. Our method avoids this problem, as seen in the Gaussian example of Section 5.3. The Diffusion-based Representation Learning method (Mittal et al., 2023) uses a separate labeling network whose output is used to guide a denoiser. The two networks are jointly trained to minimize a combination of denoising error and the KL divergence of the label distribution with a standard Normal (similar to objectives for variational AutoEncoders). Trained networks showed success in recognition, but properties of the learned conditional density were not examined. Subsequent work (Wang et al., 2023) augmented the DRL objective with an additional mutual information term. And finally, (Hudson et al., 2024) the SODA combines three networks: an image encoder, a denoiser, and a bridge network that maps the encoding into gains and offsets that are used to drive the conditioning of the denoiser. The entire model is trained on a single denoising loss, and generates images of reasonable quality, but the properties of the learned density and embedding space were not analyzed.

Guiding score diffusion with projected scores raises many questions. The embedding space of our current model relies on feature vectors constructed from channel averages. This is a natural choice of summary statistic, especially for images drawn from stationary sources. However, Figure 4 shows that many of these channels, in the first and last layers, are not providing much benefit in differentiating classes. This suggests that they could be eliminated, further reducing the dimensionality. The construction of the feature vector from alternative linear projections of channel responses may also provide a useful generalization for capturing spatially varying properties of image classes. Finally, an outstanding mathematical question is to understand the accuracy of stochastic interpolants (Albergo M., 2023) obtained with projected scores, and how it relates to feature space properties.

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

# A  ARCHITECTURE, DATASETS AND TRAINING

**Architectures.**  We use UNet architecture Ronneberger et al. (2015) that contain 3 decoder blocks, one mid-level block, and 3 decoder blocks. Each block in the encoder consists of 2 convolutional layers followed by layer normalization and a ReLU non-linearity. Each encoder block is followed by a $2 \times 2$ spatial down-sampling and a 2 fold increase in the number of channels. Each decoder block consists of 4 convolutional layers followed by layer normalization and a ReLU non-linearity. Each decoder block is followed by a $2 \times 2$ spatial upsampling and a 2 fold reduction of channels. The total number of parameters is 11 million.

The same architecture is used for the feature guided models (conditionals) as shown in Figure 9. To compute $\phi(x)$, spatial averages of the last layer's activations are computed per channel for each block. The total number of channels used in $\phi$ computation is 1344, so $\phi \in \mathbb{R}^{1344}$. A matching method is added to the code to subtract $\phi(x_\sigma)$ and add $\phi(x')$. The only change in this UNet compared to the vanilla architecture is a multiplicative gain parameter, $g$, which is optimized during training. In sampling, it is multiplied with $(\phi_y - \phi(x_\sigma))$. Note that the addition of multiplicative gain parameter only resulted in minor improvements in performance, so the feature guided model can be implemented without them.

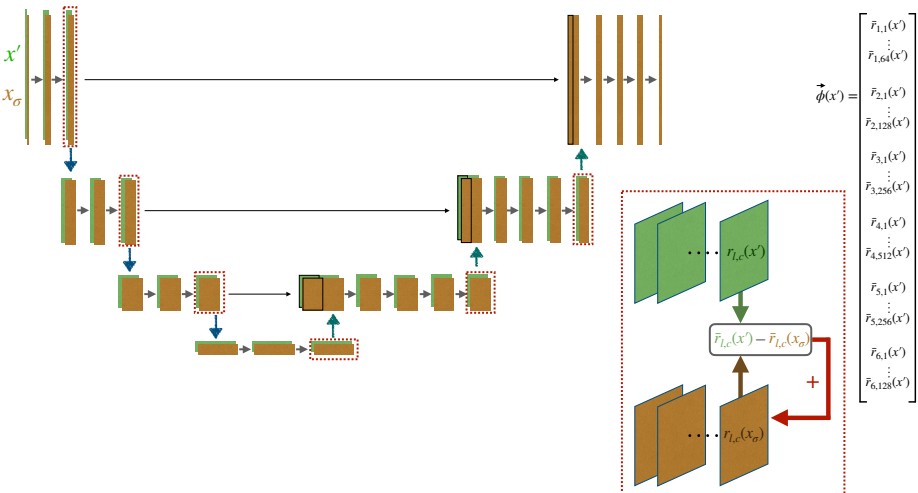

Figure 9: Conditional UNet architecture, implementing our feature-guided score $s_\theta(x_\sigma, x')$. The same network is used to compute conditioning features (green), and the denoiser (brown). Spatial averages of indicated channels (dashed red boxes) are measured from conditioning image $x'$, and imposed on the denoiser acting on $x_\sigma$ (blown up dashed red box).

**Datasets.**  The dataset contains 1700 images of $1024 \times 1024$ resolution. Each image is partitioned into non-overlapping patches of size $80 \times 80$, resulting in 144 patch per texture image or class. In each class, 140 crops are assigned to the training set and 4 crops are assigned to the test set. The total number image patches in the training set is $234,000$. The patch size was chosen intentionally to match the receptive field size of the network at the last layer of the middle block. This is to enable the network to capture global structure of the patch.

For experiment shown in Figure 3, we collected a dataset of 160 images of a single class by taking photographs of a single wood texture. The images are high resolution ($3548 \times 5322$) and are cropped to non-overlapping $80 \times 80$ patches. The total number of patches in the dataset is $125,000$. This large number of patches in the training set is required to ensure that the learned model is in the generalization regime (Kadkhodaie et al., 2024).

**Training.**  Training procedures are carried out following Algorithm 4 or Algorithm 2 by minimizing the mean squared error in denoising images corrupted by i.i.d. Gaussian noise with standard deviations drawn from the range $[0, 1]$ (relative to image intensity range $[0, 1]$). Training is carried out on batches

of size 512, for 1000 epochs. Note that all denoisers are universal and blind: they are trained to handle a range of noise, and the noise level is not provided as input. These properties are exploited by the sampling algorithms (3 and 1), which can operate without manual specification of the step size schedule.

# B    ALGORITHMS FOR LEARNING AND SAMPLING: FEATURE-GUIDED MODEL

Feature-guided score diffusion is implemented using the Stochastic Iterative Score Ascent (SISA) algorithm (see Appendix C).

In the main text, we use the notation $s(x_\sigma, \phi(x') - \phi(x_\sigma))$ to refer to the score network. However, note that $s$ and $\phi$ are implemented by the same network parameterized by $\theta$, so in practice $\phi(x_\sigma)$ is not an input argument to $s$, but is computed by $s$ from $x_\sigma$ and then used in $(x_\sigma, \phi(x') - \phi(x_\sigma))$ within the layers of the same network. We chose the notation to make it explicit that dependency of projected score on the feature vector is only through the deviation between the feature vectors. In practice, however, the network $s_\theta$ first computes $\phi(x')$ from an image or a batch of images and then operates on $x_\sigma$ while adding $\phi(x')$ and subtracting $\phi(x_\sigma)$. So to make the notation in the algorithms consistent with implementation, we write $s(x_\sigma, x')$. Algorithm 1 describes all the steps of sampling using feature guided diffusion model. The core of the algorithm is to compute the projected score, take a partial step in that direction and add noise:

$$x_{\sigma_k} = x_{\sigma_{k-1}} + hs(x_{\sigma_{k-1}}, \{x_i\}_{i \leq n}) + \gamma_k z_k$$

To compute the projected score, $\phi(x)$ and $\phi(x_\sigma)$ are computed in the same $s$ network. At each stage, $\phi(x)$ is added to and $\phi(x_\sigma)$ is subtracted from the activations. This amount to a forward pass for $x$ and a forward pass for $x_\sigma$ to compute the projected score. For more efficiency in sampling, the $\phi_y$ of the conditioning density can be stored and reused to avoid redundant computation. Note that $n$ can be set to 1 for efficiency without hurting the performance.

---

**Algorithm 1** Sampling using feature guided score diffusion

---

**Require:** data from conditioning class $\{x_i\}_{i \leq n} \in y$, projected score network $s(x_\sigma, \{x_i\}_{i \leq n})$, step size $h$, injected noise control $\beta$, initial noise $\sigma_0$, final $\sigma_\infty$, mixture distribution mean $m$
1: $k = 0$
2: Draw $x_{\sigma_0} \sim \mathcal{N}(m, \sigma_0^2 \mathrm{Id})$
3: **while** $\sigma_k \geq \sigma_\infty$ **do**
4:    $k \leftarrow k + 1$                                                                 ▷ Compute the projected score
5:    $\sigma_k^2 = \|s(x_{\sigma_{k-1}}, \{x_i\}_{i \leq n})\|^2 / d$                     ▷ Compute the current noise level
6:    $\gamma^2 = \left((1 - \beta h)^2 - (1 - h)^2\right) \sigma_k^2$
7:    Draw $z_k \sim \mathcal{N}(0, I)$
8:    $x_{\sigma_k} = x_{\sigma_{k-1}} + hs(x_{\sigma_{k-1}}, \{x_i\}_{i \leq n}) + \gamma_k z_k$     ▷ Update line with projected score
9: **end while**
10: **return** $x$

---

Algorithm 2 describes all the steps for training a projected score model. The network $s(x_\sigma, x')$ takes a pair of images. $\phi(x')$ and $\phi(x_\sigma)$ are computed using the same $s_\theta(x_\sigma, x')$ network in the forward pass and added to and subtracted from the activations respectively.

---

**Algorithm 2** Learning a projected score network

---

**Require:** data partitioned to different classes $\{x_i, y_i\}_{i \leq n}$, UNet architecture $s_\theta(x, x')$
1: **while** Not converged **do**
2:    Draw $x, x'$ of label $y$ from training set
3:    Draw $\sigma \sim \mathrm{Uniform}[0,1]$
4:    Draw $z \sim \mathcal{N}(0, \mathrm{Id})$
5:    $x_\sigma = x + \sigma z$
6:    $\nabla_\theta \|\sigma z - s_\theta(x_\sigma, x')\|^2$                                 ▷ Take a gradient step
7: **end while**
8: **return** $s = s_\theta$

---

# C  ALGORITHMS FOR LEARNING AND SAMPLING: MIXTURE MODEL

Stochatsic Iterative Score Ascent algorithm (SISA) was introduced by Kadkhodaie & Simoncelli (2020). It is an adaptive diffusion algorithm, where the time schedule is set by the model automatically using the estimated noise level at each time step. Here, for completion, we include these algorithms. For experiments which involved a mixtures model, the training and sampling were done using Algorithm 4 and Algorithm 3. We set the parameters to $h = .01$ and $\beta = .05$.

---

**Algorithm 3** Sampling with Stochastic Iterative Score Ascent (SISA)

---

**Require:** weighted score network $s_\sigma(x)$, step size $h$, injected noise control $\beta$, initial $\sigma_0$, final $\sigma_\infty$, distribution mean $m$

1: $t = 0$
2: Draw $x_0 \sim \mathcal{N}(m, \sigma_0^2 \mathrm{Id})$
3: **while** $\sigma_t \geq \sigma_\infty$ **do**
4:     $t \leftarrow t + 1$
5:     $\hat{\sigma}^2 = \|s(x_{t-1})\|^2 / d$              ▷ Approximate an upper bound on current noise level
6:     $\gamma_t^2 = \left((1 - \beta h)^2 - (1 - h)^2\right) \hat{\sigma}^2$
7:     Draw $z \sim \mathcal{N}(0, I)$
8:     $x_t = x_{t-1} + hs(x_{t-1}) + \gamma_t z$          ▷ Perform a partial denoiser step and add noise
9: **end while**
10: **return** $x_t$

---

**Algorithm 4** Learning a score network

---

**Require:** UNet architecture $s_\theta(x)$ computing a score parameterized by weights $\theta$ and weighted by $\sigma^2$. Clean images $x$.

1: **while** Not converged **do**
2:     Draw $x$ from training set
3:     Draw $\sigma \sim \text{Uniform}[0,1]$
4:     Draw $z \sim \mathcal{N}(0, \mathrm{Id})$
5:     $x_\sigma = x + \sigma z$
6:     $\nabla_\theta \|\sigma z - s_\theta(x_\sigma)\|^2$             ▷ Take a gradient step
7: **end while**
8: **return** $s = s_\theta$

---

## D    ADDITIONAL EMPIRICAL RESULTS

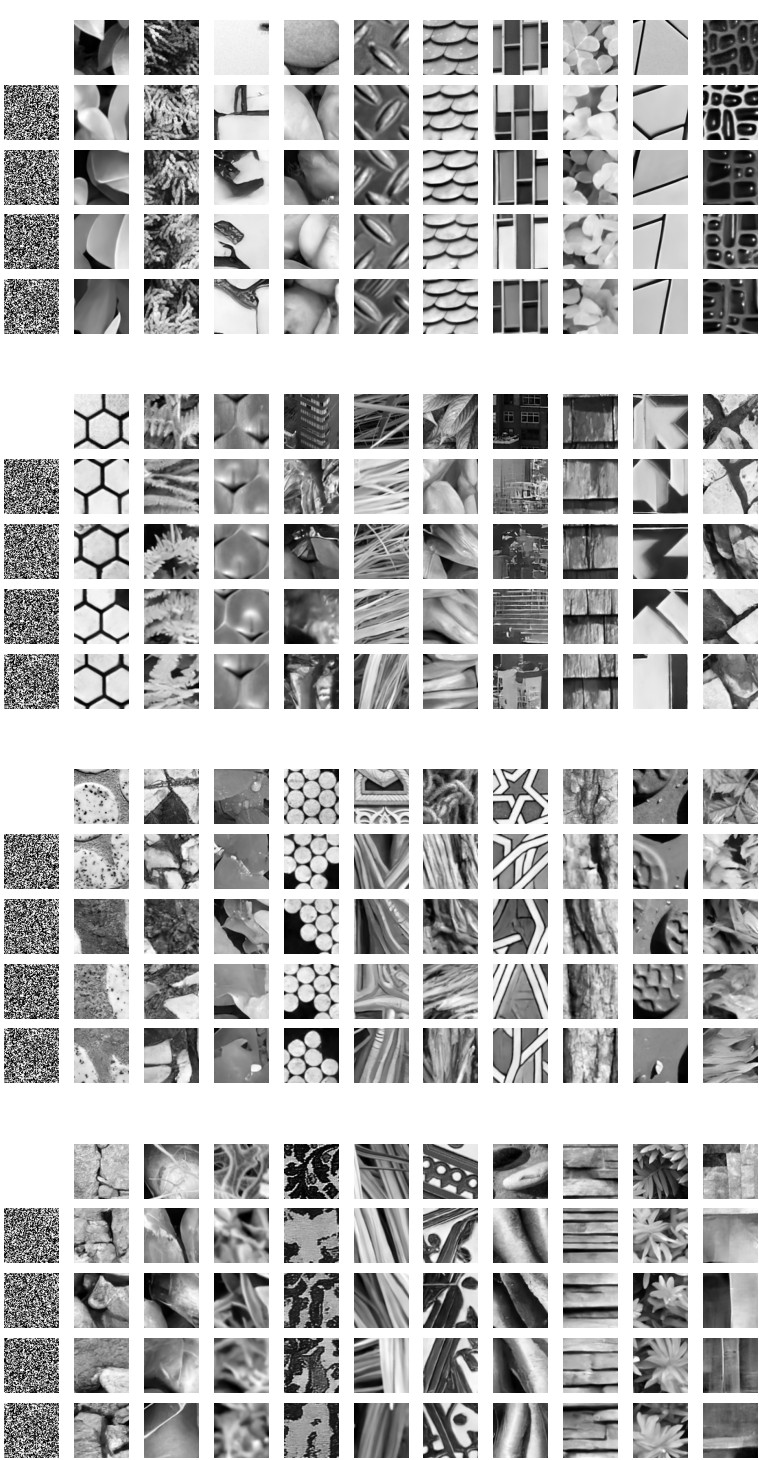

Figure 10: More examples of conditional sampling. Top row shows example images from different conditioning classes $y$. Leftmost column shows 4 initial (seed) noise images. Remaining columns show 4 images sampled using the conditional model, conditioned on the feature vector $\phi_y$ for the corresponding class. Here the class feature vector is obtained from a single image from a class (i.e., setting $n_y = 1$ in Algorithm 1). Hyperparameters in sampling algorithm are set to $h = 0.05, \beta = 0.01$

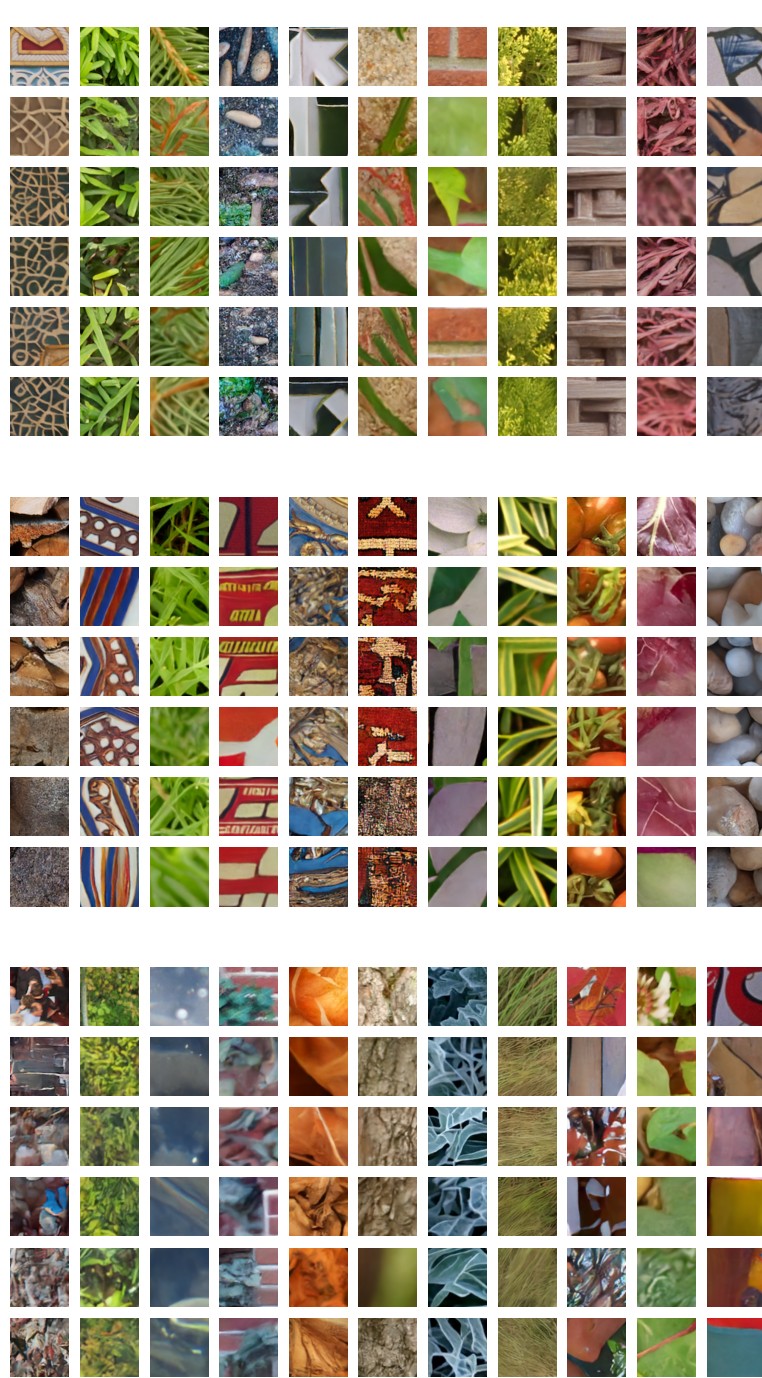

Figure 11: More examples of conditional sampling from a model trained on color texture images. Top row shows example images from different conditioning classes $y$. See caption of Figure 10.

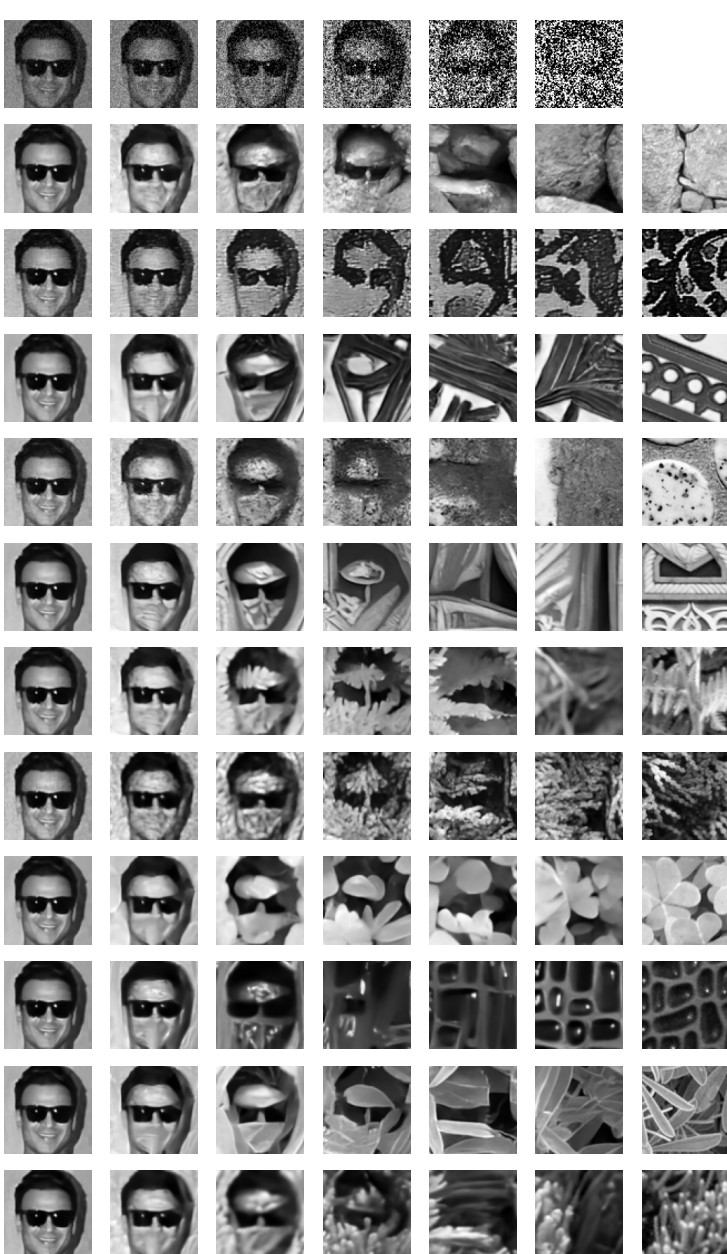

Figure 12: Effect of conditioning depends on the noise level $\sigma$. **Rightmost column:** Conditioning image from a class. **Top row:** different levels of Gaussian noise is added to a face image from the CelebA dataset Liu et al. (2015). All the other rows show conditional samples drawn starting from the initial image shown on the first row. The feature guided sampling algorithm is applied to the noisy image with conditioning on different classes. The effect of conditioning changes as a function of noise level. At smaller noise levels the effect of the conditioner is to add fine features (details) to the initial image. When the noise level is higher on the initial image, the conditioning introduces larger more global features to the final sample.

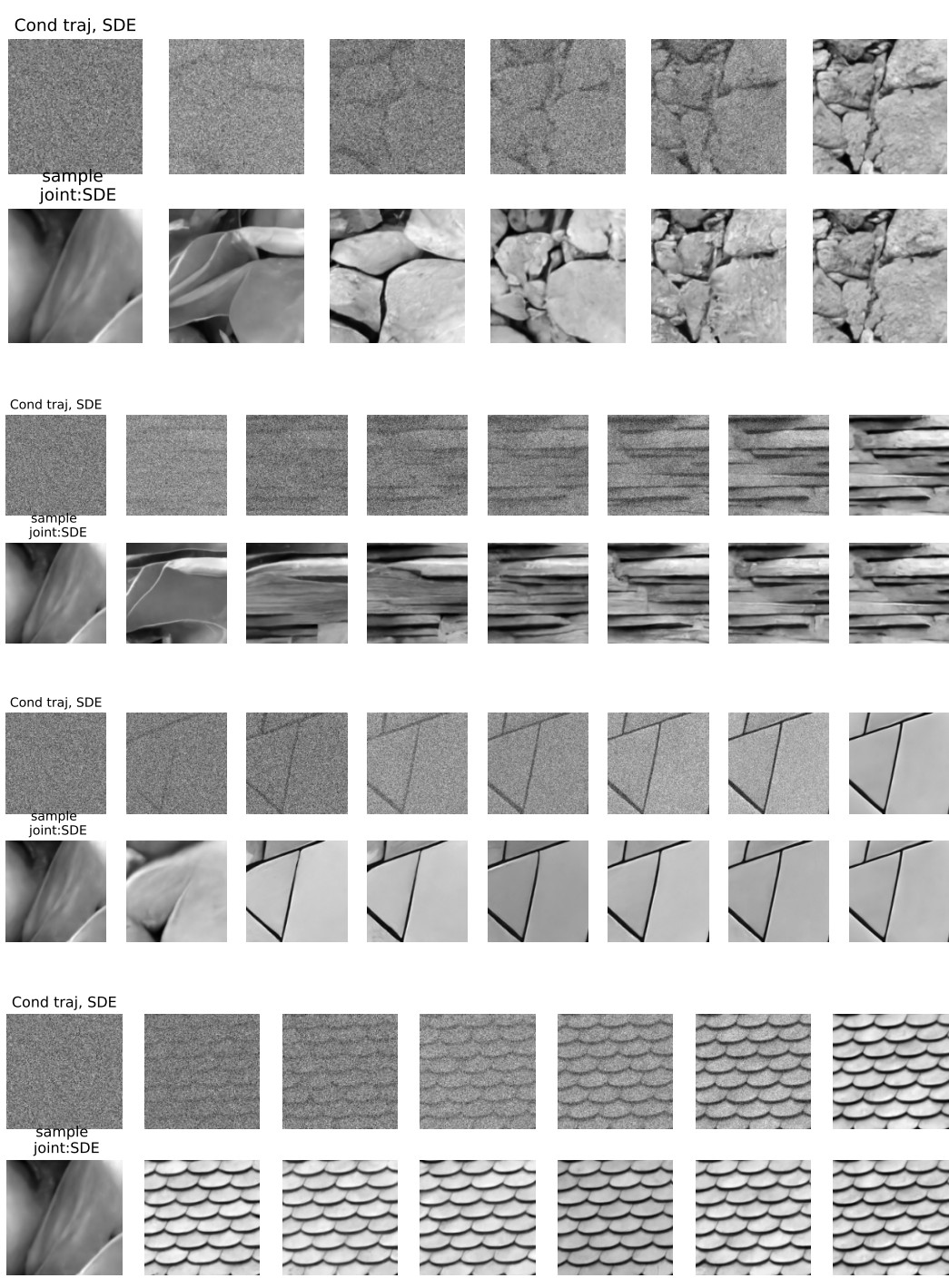

Figure 13: Effect of conditioning at different noise levels on sampling. In each of the 4 sub-figures, the top row shows a sampling trajectory using feature guided score diffusion (Algorithm 1), starting from the same sample of noise and generating an image from the conditioning class. The second row shows final samples generated without conditioning (Algorithm 4) starting from the intermediate point of the trajectory shown above it. This is akin to turning off the conditioning at an intermediate noise level. After the trajectory is within the basin of attraction of a class, shutting down conditioning does not change the sample outcome as predicted in Section 3. The exact noise level at which the trajectory becomes independent of conditioning depends on the conditioning class (and probably its distance from other classes).

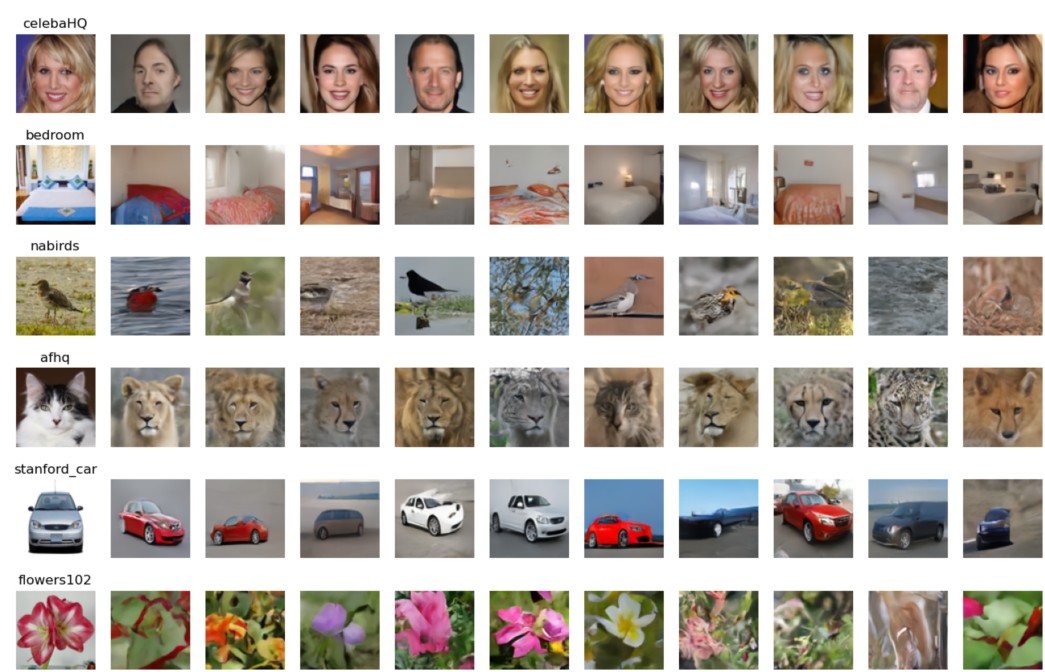

Figure 14: Uncurated samples generated from a model trained on a mixture of six common datasets down sampled to $80 \times 80$: CelebaHQ $(30k)$, a subset of LSUN bedroom class $(30k)$, AFHQ $(16k)$, Flowers102$(8k)$, Stanford cars$(16k)$, and a subset of North American Birds $(30k)$. Total number of images in the entire set is $130,000$. We use the same architecture described in Appendix A. In each row the leftmost image is an example image from the training set from the class. To obtain samples from a class, we condition on $\phi_y$ for the class computed from a batch of images. The algorithm parameters are $h = 0.01$ and $\beta = 0.01$

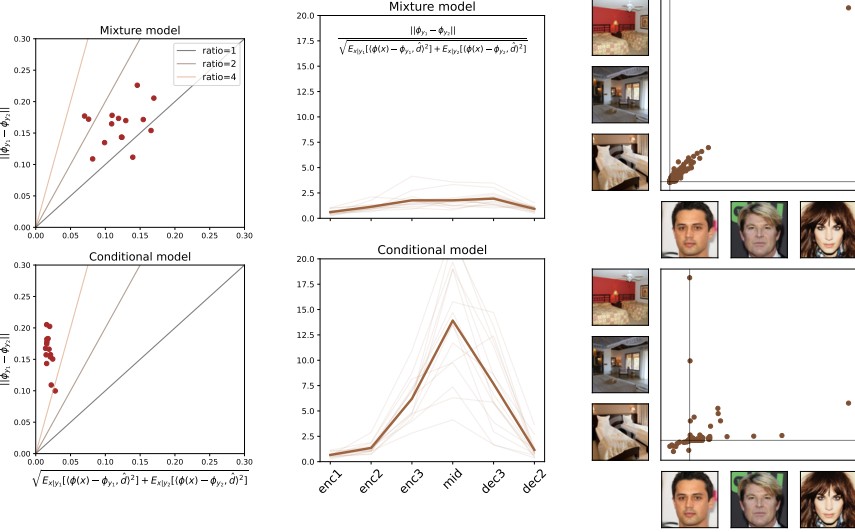

Figure 15: Concentration of image feature vectors $\phi(x)$ within class, and separation of centroids $\phi_y$ between classes for multi-class dataset. Top row shows results for the unconditioned mixture model and the bottom row shows results for the conditional model. See caption of Figure 4 for details.

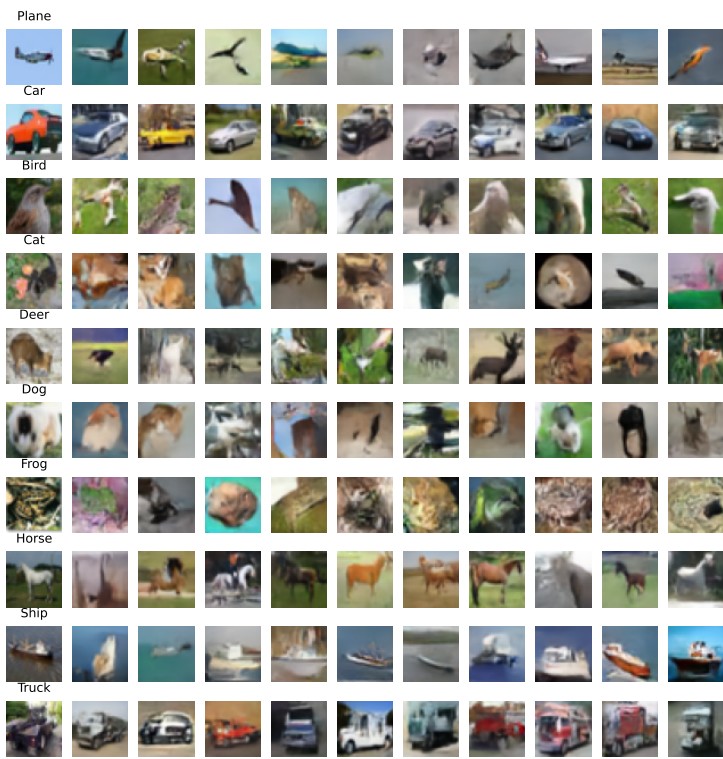

Figure 16: Uncurated samples generated from a model trained on CIFAR10 dataset. In each row the leftmost image is an example image from the training set from the class. To obtain samples from a class, we condition on $\phi_y$ for the class computed from a batch of images. The algorithm parameters are $h = 0.05$ and $\beta = 0.01$.

