# OpenReview forum: "Feature-guided score diffusion for sampling conditional densities"
_ICLR.cc/2025/Conference — Submitted to ICLR 2025_

### Official Review · Reviewer_Epc9 · 2024-10-26

**Soundness:** 1
**Presentation:** 1
**Contribution:** 1
**Rating:** 3
**Confidence:** 3

**Summary:**

This paper focuses on the problem of conditional generation with diffusion models, and specifically class-conditional generation. The authors claim that commonly used class-conditional sampling methods are inexact, and thus propose a new method for class-conditional sampling with diffusion models. Their method, taking inspiration from a GMM distribution, attempts to train a diffusion model that is biased to a certain class based on the features of some given conditioning image. The authors demonstrate several examples of their method.

**Strengths:**

- Conditional sampling is a key problem, where improvements may have a high impact.
- The proposed solution is novel, to the best of my knowledge.

**Weaknesses:**

- The paper's motivation is not sufficiently clear. While adding noise and conditioning may not commute in the general case, I believe the operation do commute in the case where each sample image has only one corresponding class, which is the common case in class-conditional sampling.
- Many conditional diffusion models in the literature are trained with the condition as an input to the network, alleviating the concern regarding the inexactness of using Eq. (4) for sampling. While such method require training a conditional model, but so does the method proposed in the paper, making the advantage here unclear.
- The experiment section is missing a meaningful comparison with alternative conditional generation methods, despite those methods being well established and mentioned in the text. Could the authors to compare their method to existing conditional diffusion models, and highlighting specific advantages or differences?
- The proposed method relies on access to an example image from a given class for sampling, making generation more cumbersome. this concern is not addressed in the paper. Perhaps this could be meaningfully discussed in a limitations section.
- Please address several typos in the discussion section.

**Questions:**

Please see concerns under Weaknesses.
- Why have the authors chosen not to use an established class conditional dataset, such as CIFAR10 or ImageNet for the experiments in the paper?
- Can the authors elaborate why adding noise and conditioning do not commute? I believe such a claim requires proof.

---

> ### Author Response · Authors · 2024-11-22
>
> Thank you for your review which will help us improve the paper. We provide answers to each of your concerns below, and have updated the paper accordingly.
>
> **Weaknesses:**
> 1. We agree that the paper motivation was not sufficiently clear. We have now re-written the conditional sampling subsection to clarify our motivation (see lines 125 to 145 in the updated paper). We removed the phrase about “conditioning and adding noise not commuting”. Although the statement is correct, we realized the presentation was not immediately clear. The argument in the paper is now described as : “In practice, one might employ neural network classifiers trained on clean data to estimate the likelihood. This however introduces an error because $p_\sigma (y|x_\sigma ) \neq p(y|x_\sigma )$. In other words, the correct likelihood of y for noisy data cannot be computed by evaluating the likelihood function $p(y|.)$ on noisy data. The likelihood function changes at different noise levels.” This is a more elaborated version of “adding noise and conditioning do not commute”. This is simply to say the Bayes rule (eq4) holds if one first adds noise and then conditions, but it does not hold if one first conditions and then adds noise. Specifically:
> - Adding noise and conditioning: convolving $p(x)$ with Gaussian kernel  $g$ and then conditioning results in $p_\sigma(x_\sigma |c)$
> - Conditioning and adding noise: convolving $p(x|c)$ with a Gaussian kernel results in $ g(z) * p(x|c) = g(z) * (p(c|x)p(x)/p(c))$
> - The two are not equal. Please also see the first page of Chidambaram et. al, 2024 (cited in our paper) for more details.
>
> 2. It is true that many conditional diffusion models in the literature are trained with the condition as an input, but existing algorithms do not provide samples from the appropriate conditional distributions, as demonstrated in the literature that we cite in the Background section. This includes classifier-free guidance methods where the network is trained conditionally.
> Several publications (cited in our submission) have shown that the choice of weighting that leads to higher quality samples achieves this at the expense of lower diversity, for example for the mixture of two Gaussians. We show that our method does not fail for mixture of two Gaussians. At the same time, it succeeds in returning visually high quality samples of complex images (see next point).
>
> 3. To address comparisons over datasets used by other algorithms, we generated new results. We applied our method to the CIFAR dataset of small images and also to larger size images in a dataset constructed as a union of 6 commonly used datasets. Results are presented in Figures 14 to 16 (AppendixD). We used the same architecture for both of these experiments: a UNet with 3 encoder and decoder bocks. Despite the increased complexity of these datasets relative to the texture example in the original submission, we find that the model succeeds in drawing visually high quality samples from the correct class. For quantitative comparison we need to train on Imagenet dataset which is a substantially larger and more complex dataset, with a relatively small number of samples in each class.  We expect we would need to increase the size of the network and modify the architecture by adding commonly used elements such as attention blocks, BigGANs down-sampling blocks, etc. This is a worthy endeavor, but not one we could attempt during the rebuttal period.
>
> 4. Our method does not “rely on access to an example image for a given class for sampling”. During synthesis, it only needs access to $\phi_y$ which is a pre-computed feature vector of the class. So each iteration involves only one forward pass and it is not cumbersome.
> During training, we rely on access to pairs of images from the same class. So each training iteration involves 2 forward passes. We will clarify this important distinction in the revised paper
>
> 5. The typos in the discussion section have been fixed.
>
> **Questions:**
> - Our motivation for choosing a texture dataset was to examine a *continuous* conditioning space (as opposed to discrete object classes), and to sidestep any requirement of labeled data.  In this dataset, with 1700 classes and only 140 images per class, we find that a relatively small and simple fully convolutional UNet network succeeds in learning the conditional densities. Achieving the same success for complex and discrete object classes (such as those in ImageNet), will require much larger networks and  datasets with many images per class. However, as discussed above, we now included two new experiments on more common datasets.
>
> - See first reply under “Weaknesses”
>
> Regarding the summary: in case there is any misunderstanding:  the method we propose does not rely on a GMM assumption. The GMM example is provided as a numerical demonstration that we can sample from the correct conditional distribution, which has been shown to fail for previous guidance algorithms.

---

> > ### Comment · Reviewer_Epc9 · 2024-11-26
> >
> > I thank the reviewers for their response. Nevertheless, I believe that most of my concerns have yet to be addressed, and thus I retain my original score. Specifically, I find that the authors must demonstrate an advantage over simple class-conditional diffusion models (which remains exact according to the authors when not used with CFG), to motivate this alternative approach. Similarly, some qualitative comparisons are crucial in my opinion, even on small datasets such as CIFAR10.

---

> > > ### Author Response · Authors · 2024-11-27
> > >
> > > Our method has two important advantages which we tried to elaborate in our response. We reiterate a more concise version here:
> > >
> > > 1. Previous method (both classifier-guidance and classifier-free guidance) have a trade-off between high quality samples vs sampling from the correct conditional density. Specifically when $\omega = 1$ they sample from the correct density but the quality is poor. When $\omega \neq 1 $ the sample quality is good but they do not sample from the correct density.
> > > Our method **does not exhibit this trade-off**: It samples from the correct conditional in the tractable case of a Gaussian (fig 6) AND it returns high quality samples in the case of real images (original figs 7, 11, and new figs 14, 16).
> > >
> > > 2. Additionally, our method is the first to **learn an embedding space within a denoising diffusion model**. One of the main disadvantages of diffusion models is that unlike VAEs they do not produce an explicit latent space representation. This is particularly important for scientific application where understanding the implicit learned density matters. We show the effectiveness of the learned representation by showing that it successfully guides the conditional sampling. The success is because the classes are concentrated in the embedding space (fig 4 and 15) and Euclidean (fig 5).
> > >
> > > We believe these contributions - learning and using a representation in diffusion models - are significant and aligned with the topic of ICLR.
> > >
> > > Also, please note that our updated paper includes conditional samples from our model trained on CIFAR10 (fig16).
> > >
> > > We are wondering which of your concerns have not been addressed? We can provide more details.

---

### Official Review · Reviewer_hvBa · 2024-11-02

**Soundness:** 2
**Presentation:** 3
**Contribution:** 2
**Rating:** 5
**Confidence:** 4

**Summary:**

The most common approach to guide diffusion models is by using score guidance, whether in the form of classifier guidance or in the form of classifier free guidance. The paper claims that this approach relies on an approximation and is thus inaccurate. As an alternative, it proposes a simple method for training a conditional diffusion model. The approach is motivated intuitively and tested empirically.

**Strengths:**

- Classifier free guidance (CFG) is the most dominant approach for guiding diffusion models today, even though it is known to lead to biased densities. Several recent papers analyzed the drawbacks of the approach from a theoretical standpoint. However the topic of designing good practical alternatives to CFG is still under-explored. This paper attempts to fill this gap, which is undoubtedly an important goal.
- The paper presents clear intuition and empirically validates that the assumptions underlying the proposed construction hold.

**Weaknesses:**

- The whole motivation of the paper is to propose an alternative to existing guidance methods. However, it does not provide theoretical guarantees that the approach samples from the conditional distribution. And it also does not provide any empirical evidence that the proposed approach outperforms the standard way of conditioning diffusion models. Specifically, it does not compare the sampling quality to that obtained with a conditional denoiser (with the common conditioning mechanism for U-Nets), neither without nor with CFG. It also does not present any quantitative measure of sample quality (e.g. FID) on natural images, and does not present results on popular datasets like CIFAR or ImageNet, which prevents from evaluating whether the proposed method leads to SOTA results. As such, it is impossible to evaluate the effectiveness of the proposed method.
- I believe that the motivation presented in the paper is inaccurate and somewhat misleading. Using score guidance (as in Eq. (4)) should theoretically lead to accurate results. What is inaccurate in CFG is that when taking the guidance parameter w to be greater than 1, it does not sample from the titled distribution p(x)p(y|x)^(1+w) from which it attempts to sample, but rather from a different distribution. See for instance the closed form expressions for the Gaussian setting in [1] (Eqs. (11),(12)), where the distributions of samples obtained with CFG-DDPM and CFG-DDIM are incorrect only when the CFG parameter (gamma) is greater than 1, but are correct when it equals 1. In particular, the statement in L147-148 that Chidambaram et al. (2024) proved the inaccuracy of guidance on Gaussian mixtures is incorrect. They proved the inaccuracy of CFG only when the guidance parameter w is sufficiently large. When w=1 the process is theoretically accurate.


[1] Bradley and Nakkiran, "Classifier-Free Guidance is a Predictor-Corrector", arxiv 2408.09000.

**Questions:**

- Can the authors comment on how the method compares to regular guidance on common datasets, like CIFAR-10, in terms of FID?
- Can the authors comment on the motivation for suggesting an alternative to regular guidance (second weakness stated above)?
- L216: Why are the activation layer averages close to the first principal components of the network channels? Can the authors provide a proof or a reference to a paper showing this?

Typos:
- L216: principle -> principal
- L360: "some some"
- L429: missing parentheses around (m2-m1)
- L517: remove "And most"
- L518: mixeed -> mixed

---

> ### Author Response · Authors · 2024-11-22
>
> Thank you for your important remarks, which will help us to improve the clarity of the presentation.
>
> **Weaknesses:**
>
> Regarding the second point: We have updated the conditional sampling subsection to clarify our motivation. Below, we summarize the main points (see also lines 125 to 145 in the updated paper):
>
> - Classifier-guided diffusion models were motivated by Eq. 4. In the context of diffusion models, it is necessary for eq 4 to hold at *all* noise levels. As a result, the correct likelihood at all noise levels is needed to obtain the conditional score.
> - In practice, learning the likelihood correctly from data for all noise levels has proven to be challenging, which is reflected in poor quality of samples. This motivated the idea of weighting the terms, which also appears in the classifier-free guidance methods.
> - Classifier-free diffusion is equivalent to Eq. 4 with no weighting. Hence in theory, the score is equivalent to the conditional score. However, in practice that leads to low quality samples. As a result, \omega is treated as a hyper-parameter and is often set to a value greater than one to improve quality at the expense of diversity.
> - Our goal is to introduce an alternative strategy that does not require a trading off quality with diversity. This is achieved by relying on a feature space with properties shown in the paper: separation between classes, concentration within class, and Euclidean distance metric between classes. We show that for theoretically tractable datasets (like a mixture of Gaussians), this procedure can generate samples from the correct conditional. For real data, we show that samples are visually high quality and diverse. Note that this is different from previous work, where the algorithms which sample from correct Gaussian do not lead to high quality results, while the algorithms which return high quality samples do not sample from the correct conditional Gaussian.
> - We acknowledge that for some applications, such as image generation, this tradeoff may not be so detrimental. However, for some applications (e.g. applied sciences) it can be critical not to reduce the entropy of the learned distribution. We believe our approach can be beneficial for such applications.
>
> Regarding the first point: You are right that we do not provide theoretical guarantees, and this is an important goal for future work. However, in Sections 3 and 4 we lay out a theoretical framework to analyze the dynamics of the algorithms. We also demonstrate empirically that our algorithm samples correctly from conditionals in a Gaussian mixture, and generates high quality samples from real data. We argued that this success is due to the richness of the feature space which is learned simultaneously within the same score network.
>
> **Questions:**
> - We have now generated results from two more complex image datasets (Appendix D, figures 14 to 16).  We used the same architecture as before (a UNet with 3 encoder and decoder bocks). We show that the model succeeds in drawing visually high quality samples from the correct class. Achieving SOTA on FID is a worthy goal, but the purpose of the current work is to demonstrate that we can learn a feature space, computed within the score network and using only the denoising objective, that supports effective conditional sampling. Reporting FID scores would require training our model on the Imagenet dataset. For that, we would need to increase the size of the architecture and include elements like attention blocks, BigGAN downsampling and other elements. This is not feasible given the time constraints of the rebuttal period, but we consider it a natural future direction for the work.
>
> - As elaborated above, the motivation for introducing this alternative method is to avoid the trade-off between diversity and quality that is found in previous methods. Moreover, our results indicate that the learned feature embedding captures interesting properties of the representation within the score network. Given the complexity of diffusion models, we think it is valuable to have a simple working model which consists of just one network which lends itself to more analysis. This can be an avenue for better understanding the inner workings of diffusion models.
>
> - For uncentered data (which is the case here) the correlation matrix depends upon  the mean. For positive images, especially when they are stationary, the largest variance is often in the direction of the mean, which is thus the dominant principal component. Although this is not mathematically guaranteed (one can define or construct a positive random process whose correlation matrix has a principal component not aligned with the mean), it can be verified for our neural network channels.
>
> Thanks for catching these typos, which we’ve fixed. Also, thanks for mentioning the Bradley & Nakkiran paper. We were not aware of this work, which is relevant to our motivation. We have included it in the background  section.

---

> > ### Comment · Reviewer_hvBa · 2024-11-26
> >
> > I thank the authors for the effort in providing additional experiments. However, I still believe that the proposed method has bot been fully evaluated to convince that it is beneficial over training a conditional denoiser (even without a classifier-free guidance parameter that is greater than 1). I would expect to see FID comparisons at least with this simple baseline on CIFAR. I understand that the approach may not lead to state-of-the-art FID scores. That's completely fine, as long as an apples-to-apples comparison is provided: Taking the same network architecture and training regime, once with a standard conditioning mechanism, and once with the proposed method.
> >
> > Considering the paper's current state, I choose to keep my score.

---

### Official Review · Reviewer_wJvb · 2024-11-04

**Soundness:** 3
**Presentation:** 3
**Contribution:** 3
**Rating:** 8
**Confidence:** 3

**Summary:**

This paper introduces a guided score diffusion method that samples from class conditional distributions by calculating a projected score based on feature vectors, avoiding the need to directly estimate the scores of those densities. The method employs a neural network trained to minimize a single denoising loss, with the feature vector defined as spatial averages from selected layers, leading to a Euclidean embedding of class conditional probabilities. It effectively clusters learned features around their centroids, allowing for accurate sampling from target conditional distributions and facilitating smooth transitions between classes through linear combinations of mean feature vectors.

**Strengths:**

This is a well-written paper. Both score and feature vectors are represented with the same network. The learned feature vectors cluster around their centroids, which enhances the accuracy of sampling rom the conditional probability density. The method enables gradual transitions of the images between classes through linear interpolation of mean feature vectors. The experimental results show that a diffusion algoriothm based on the projected score provides an accurate sampling of conditional probabilities.

**Weaknesses:**

The authors provided a way to build the feature vectors that share the same network weights as the score function. It is not clear how to determine the feature vector dimension.

**Questions:**

How does the feature dimension affects the learning and generation?

---

> ### Author Response · Authors · 2024-11-22
>
> Thank you for your encouraging comments and constructive questions.
>
> **Weaknesses/Questions:**
> In our implementation, the feature vector is constructed from spatial averages of channels in the last layers of each of the U-Net blocks. Our UNet has the following architecture:
> - 3 encoder blocks with {64,128,256} channels, respectively.
> - middle block with 512 channels.
> - 3 decoder blocks with {256,128,64} channels, respectively.
>
> This gives a feature vector with a total of 1344 dimensions (please see new architecture diagram in updated appendix A).
>
> The middle column of Figure 4 (now updated) suggests that we might be able to reduce the dimensionality of the feature space by eliminating some of the early and late channels, which contribute less to the separation of classes.  More generally, we are interested in reducing to a minimal dimensionality, and in understanding the relationship between this minimal dimensionality and the complexity of the data distribution.  The feature vector must be sufficiently rich in order to separate elements of different classes. With our current experiments, overparameterization of the feature vector does not seem to affect the learning or generation negatively.

---

> > ### Comment · Reviewer_wJvb · 2024-11-27
> >
> > Thank you to the authors for their detailed responses and additional results. The authors have addressed my questions. Classifier-Free Guidance (CFG) is a widely used method for guiding diffusion models, but it can result in biased densities and may require a trade-off between diversity and quality. This paper introduced an innovative way to address those challenges, which I think is an important contribution.  The mixture of Gaussians and synthetic examples clearly show the advantage of the proposed method. I maintain my current score.

---

### Official Review · Reviewer_9mDa · 2024-11-04

**Soundness:** 4
**Presentation:** 3
**Contribution:** 3
**Rating:** 6
**Confidence:** 4

**Summary:**

This paper introduces a novel guided score-based diffusion model that does not require any additional structural modifications. Instead, it extracts features from key layers of its own score estimation model (i.e. Unet) and applies spatial averaging, enabling guidance control in the feature space. Unlike methods such as classifier-free guidance, this feature-guided diffusion does not rely on likelihood estimation, allowing for a more accurate estimation of conditional density.

**Strengths:**

1. The paper is well-structured, easy to read, and highly innovative, introducing a projected score embedded in feature space. This embedding is not only straightforward to obtain (directly extracted from the score estimation model) but also adheres to Euclidean interpolation properties.

2. The model successfully achieves conditional generation in a mixture of Gaussian distributions, demonstrating that the feature-guided score diffusion model can accurately capture conditional density—an ability lacking in many other approaches.

3. Compared to mixture models, the feature-guided score diffusion model shows stronger concentration and separation across different classes within the embedding space.

**Weaknesses:**

The dataset used in experiments is overly simple. The training dataset is derived by cropping 1700 images into 234k patches. Although the patches are non-overlapping, the data distribution for each class lacks sufficient diversity. Experiments on a more complex dataset, like ImageNet, would strengthen the paper’s validity.

**Questions:**

1. What is the difference between feature space and latent space in stable diffusion? While the latter requires an additional encoder, how does the former feed $x$ and $x'$ into the U-Net?

2. In Fig. 3 (left), the feature-guided model shows strong performance across all noise levels. Does this suggest that fewer NFE (Number of Function Evaluations) could be used during conditional generation? Designing experiments to verify this could strengthen the paper.

3. In Fig. 4 (middle), were the two metrics normalized? Otherwise, this comparison might not be on the same scale. Additionally, the variance in image feature vectors appears large—is this due to insufficient training? Please clarify further.

---

> ### Author Response · Authors · 2024-11-22
>
> Thank  you for your constructive comments and questions. We have updated the paper, and refer below to the specific changes.
>
>
> **Weaknesses**:
>
> 1. We agree that testing our method on a more complex dataset would strengthen the paper. Following the suggestion of reviewer hvBa, we have first applied our method  CIFAR dataset of small images - results are shown in figure 16 (Appendix D) of the updated paper.   We have also applied it to larger images in a dataset constructed from the union of 6 commonly used datasets: CelebaHQ (30k images), a subset of LSUN bedroom class (30k), AFHQ (16k), Flowers102(8k), Stanford cars(16k), and a subset of NA_birds (30k). These results are now shown in figure 14 and 15 (Appendix D). Note that the architecture in these experiments is unchanged from the original submission: a UNet with 3 encoder and decoder blocks.  In both of these experiments, the model succeeds in drawing samples from the correct class.
> 2. Regarding the ImageNet dataset: this is a much larger dataset, with more complex images, but with a relatively small number of images (~1k) in each class.  As such, we would need to increase the size of our network substantially, and consider addition of additional architectural features such as attention blocks, BigGANs down-sampling blocks, etc.  We’ve not had sufficient time for this during the rebuttal period.
>
> **Questions**:
>
> 1. We have added a new diagram of our model architecture, illustrating how $x$ and $x’$ are fed into the network (see appendix A). Note a primary feature of our method is that there is no additional encoder - the model consists of a single UNet. Comparing our feature space to the latent space of stable diffusion: 1) Our features are extracted from the score network itself (means of channels), which are then imposed within the same score network operating on a noisy input, whereas stable diffusion uses a separate encoder to map the image to a latent space in which the diffusion/denoising takes place, and 2) Our model is trained on a single denoising objective, whereas in stable diffusion the features are extracted according to a reconstruction loss function (or some other variation). Our construction gives rise to an important property of the feature space: it defines a Euclidean metric for the conditional distributions. For example, two conditional densities which are far in terms of symmetrized KL divergence have feature vectors that are far in Euclidean distance. Other semantic differences: our feature space is a multi-scale representation so it can capture differences between class features at different scales. Overall, this feature space is quite rich, and we believe offers interesting opportunities for further study.
> 2. The suggestion based on Fig. 3  is an interesting one. We do have initial results suggesting that conditional sampling can be done in fewer steps, but these need to be verified/elaborated with additional experiments.
> 3. We’ve re-built figure 4. In particular, the middle panel now shows the ratio of the between-class distances and the within-class standard deviations, which is the relevant statistical decision theory quantity for separation of the classes.  Both plots are shown with the same units, and (as before) we show plots for pairs of classes (thin lines), and the average over all pairs.  The variability across pairs is not due to insufficient training, but arises naturally from the variable similarity between classes.

---

> > ### Comment · Reviewer_9mDa · 2024-11-26
> >
> > Thank you to the authors for their detailed responses and additional experiments, which have addressed my questions. However, I would like to retain my original score, as Figure 16 highlights the limited performance on CIFAR-10, a dataset with a more complex distribution (like Dog, Deer, Cat, Plane). My concerns regarding the practicality of feature guidance remain.

---

### Author Response · Authors · 2024-11-29
**Global comment**

While all reviewers appreciated the novelty of our work, the unresolved criticism by three reviewers was a lack of precise comparison with SOTA on sample quality (FID). We acknowledge that we did not present FID scores and that scaling up the model to do so is a valuable future direction. However, contributions of this work are aimed at other facets of the problem, orthogonal to improving SOTA on sample quality. More precisely, we demonstrated **two advantages** of our method over previous work, which are important in scientific applications:
1. Previous method (both classifier-guidance and classifier-free guidance) have a trade-off between high quality samples vs sampling from the correct conditional density (reduced diversity). Specifically when $\omega = 1$ they sample from the correct density but the quality is poor. When $\omega \neq 1 $ the sample quality is good but they do not sample from the correct density. **Our method does not exhibit this trade-off**: It samples from the correct conditional in the tractable case of a Gaussian (fig 6) AND it returns high quality samples in the case of real images (figs 7, 11, 14, 16).
2. Additionally, our method is the first to **learn and use an embedding space within a denoising diffusion model**. One of the main disadvantages of diffusion models is that unlike VAEs they do not produce an explicit latent space representation. This is particularly important for scientific application where understanding the implicit learned density matters. We show the effectiveness of the learned representation by showing that it successfully guides the conditional sampling. The success is because the classes are concentrated and separated in the embedding space (fig 4 and 15) and define a Euclidean metric such that densities closer in KL divergence correspond to closer embeddings in feature space  (fig 5).

We believe these contributions - learning and using a representation in diffusion models - are significant and aligned with the topic of the International Conference on Learning Representations.

---

### Meta-Review · Area_Chair_pLh8 · 2024-12-21

**Metareview:**

The authors present a method for score-based diffusion that doesn't depend on score guidance approximations. The four reviewers agreed that the problem is very relevant to current research directions and addresses a known issue in the field. The reviews were split two positive and two negative, and the paper overall was borderline, but below threshold for acceptance. In general, while the reviewers appreciate the relevance of the proposal, it wasn't clear to them that this particular way of addressing it provides a significant enough advance for ICLR. Issues were raised on the motivation and advantage over conditional generative methods. It was felt that the experiments did not provide enough compelling evidence in favor of the method as a practical alternative. The reviewers all engaged with the rebuttal, but three of four did not feel that their concerns were alleviated.

**Additional Comments On Reviewer Discussion:**

All reviewers all engaged with the rebuttal and chose to maintain their scores. Three of four reviewers, including one positive reviewer, did not feel that their concerns were addressed sufficiently by the rebuttal.

---

### Decision · Program_Chairs · 2025-01-22

Reject